# Computational and neural mechanisms of statistical pain learning

Flavia Mancini [1] ✉, Suyi Zhang[2] & Ben Seymour [2,3]

Pain invariably changes over time. These fluctuations contain statistical regularities which, in theory, could be learned by the brain to generate expectations and control responses. We demonstrate that humans learn to extract these regularities and explicitly predict the likelihood of forthcoming pain intensities in a manner consistent with optimal Bayesian inference with dynamic update of beliefs. Healthy participants received probabilistic, volatile sequences of low and high-intensity electrical stimuli to the hand during brain fMRI. The inferred frequency of pain correlated with activity in sensorimotor cortical regions and dorsal striatum, whereas the uncertainty of these inferences was encoded in the right superior parietal cortex. Unexpected changes in stimulus frequencies drove the update of internal models by engaging premotor, prefrontal and posterior parietal regions. This study extends our understanding of sensory processing of pain to include the generation of Bayesian internal models of the temporal statistics of pain.

The main function of the pain system is to minimise harm and, to achieve this goal, it needs to learn to predict forthcoming pain. To date, this has been studied using cue-based paradigms, in which a learned or given cue, such as a visual image, contains the relevant information about an upcoming pain stimulus[1–5]. A much more general, although neglected, route to generate predictions relates to the background statistics of pain over time, i.e. the underlying base-rate of getting pain, and of different pain intensities, at any one moment. This is important because clinical pain typically involves long-lasting streams of noxious signals, characterised by temporal regularities that underscore the temporal evolution of pain[6].

The brain can use associative learning strategies to predict pain from cues, but these algorithms do not learn the structure of the environment[7]. In principle, the pain system should be able to generate predictions by learning regularities (i.e. structures) in its temporal evolution, in absence of other information. This possibility is suggested by research in other sensory domains, showing that the temporal statistics of sequences of inputs are learned and inferred through experience - a process termed temporal statistical learning[8–13]. We hypothesise that temporal statistical learning also occurs in the pain system, allowing the brain to infer the prospective likelihood of pain by keeping track of ongoing temporal statistics and patterns.

Here, we tested this hypothesis by designing a temporal statistical learning paradigm involving long, probabilistic sequences of noxious stimuli of low and high intensities, whose transition probabilities could suddenly change. We tested people's ability to generate explicit predictions about the probability of forthcoming pain, defined the underlying computational principles and revealed their neural correlates. We investigated the following computational features: (1) is the stimulus sequence learnt using an optimal Bayesian inference strategy, or an heuristic (a model-free delta rule)? (2) Does the learning update take into account the volatility of the sequence? (3) Which temporal statistics is inferred, stimulus frequencies or transition probabilities?

After identifying the computational principles of learning to predict pain sequences, we reveal the brain regions that encode these predictions, their uncertainty and update, using functional MRI. We hypothesised statistical predictions for pain would follow the fundamental rules of optimal Bayesian inference, based on previous work on other sensory modalities[14]. We were particularly interested in understanding whether probabilistic predictions of pain might be encoded in somatosensory processing regions (primary/secondary

[1]Department of Engineering, University of Cambridge, Trumpington Street, Cambridge CB2 1PZ, UK. [2]Wellcome Centre for Integrative Neuroimaging, John Radcliffe Hospital, Headington, Oxford OX3 9DU, UK. [3]Center for Information and Neural Networks (CiNet), 1-4 Yamadaoka, Suita City, Osaka 565-0871, Japan. ✉e-mail: fm456@cam.ac.uk

somatosensory cortex and insula), given that statistical inferences of visual and auditory inputs can be encoded in visual and auditory regions[9]. This would allow us to map core regions of the pain system to specific functional information processing operations, i.e. the statistical inference of pain.

## Results

Thirty-five participants (17 females; mean age 27.4 years old; age range 18–45 years) completed an experiment with concurrent whole-brain fMRI. They received continuous sequences of low- and high-intensity electrical stimuli, eliciting painful sensations. Participants were required to intermittently judge the likelihood that the next stimulus was of high versus low intensity, given the previous stimulus (Fig. 1a, b).

We designed the task such that the statistics of the sequence could occasionally and suddenly change (i.e. they were volatile), which meant that the sequences of 1500 stimuli included sub-sequences of stimuli (mean 25 ± 4 stimuli per sub-sequence). Participants were not explicitly informed when these changes happened. Figure 1a illustrates an example of a snapshot of a typical sequence, showing a couple of 'jump' points where the probabilities change. The sequence statistics were Markovian and, thus, incorporated two types of information. First, they varied in terms of the relative frequency of high and low-intensity stimuli (from 15 to 85%), to test whether the frequency statistics can be learned. Second, sequences also contained an additional aspect of predictability, in which the conditional probability of a stimulus depended on the identity of the previous stimulus (i.e. the transition probability of high or low pain following a high pain stimulus, and the transition probability of high or low pain following a low pain stimulus; Fig. 1c). By having different transition probabilities between high and low stimuli within sub-sequences, it is possible to make a more accurate prediction of a forthcoming stimulus intensity over-and-above simply learning the general background statistics. Thus, we were able to test whether participants learn the frequency or the transition probabilities between different intensities, as shown previously with visual stimuli[14]. For this reason, our design mirrored a well-studied task used to probe statistical learning with visual stimuli[14,15]. From a mathematical point of view, the frequency can always be derived from the transition probabilities, but not vice versa. Therefore, participants were not asked to rate the frequency of the stimuli because it can be simply derived from their transition probability ratings; in contrast, transition probability estimates cannot be derived from frequency estimates.

### Behavioural results

Participants were able to successfully learn to predict the intensity (high versus low) of the upcoming painful stimulus within the sequence based on its frequency. We measured the linear relation between the true and rated frequency of low and high pain respectively, for each participant. As shown in Fig. 2a, 83% of participants showed a positive association, greater than 0, between true and rated frequencies (median Pearson's r = 0.25, SD = 0.19). Across subjects, the within-individual Pearson's r between true and rated frequencies was significantly above zero (t(34) = 6.10, p < 0.001, Cohen's d = 1.03; 2c). This indicates that the majority of participants were able to predict the frequency of the stimuli, despite the volatility of the sequence.

Next, we checked whether participants were able to also predict higher order statistics, i.e. the transition probability between the stimuli. This seemed to be quite challenging for our participants. In 74% of participants there was a positive correlation between true and rated transition probabilities (P(H|H): median Pearson's r = 0.16, SD = 0.23; P(H|L): median Pearson's r = 0.13, SD = 0.22). As shown in Fig. 2b, c, 26% of participants showed negative linear relations, indicating that they could not predict transition probabilities. At group level, the relation between true and rated transition probabilities was significantly greater than 0 (p(H|H): t(34) = 3.65, p < 0.001, Cohen's d = 0.616; p(H|L): t(34) = 3.15, p = 0.0034, Cohen's d = 0.532; note that p(H|L) and p(L|L) are reciprocal, as well as p(H|L) and p(L|L)).

Furthermore, we found no evidence for a correlation between the participant prediction accuracy (as measured by the correlation coefficient between generative and rated probabilities) and perceived pain intensity for the high pain stimulus, averaged across sessions

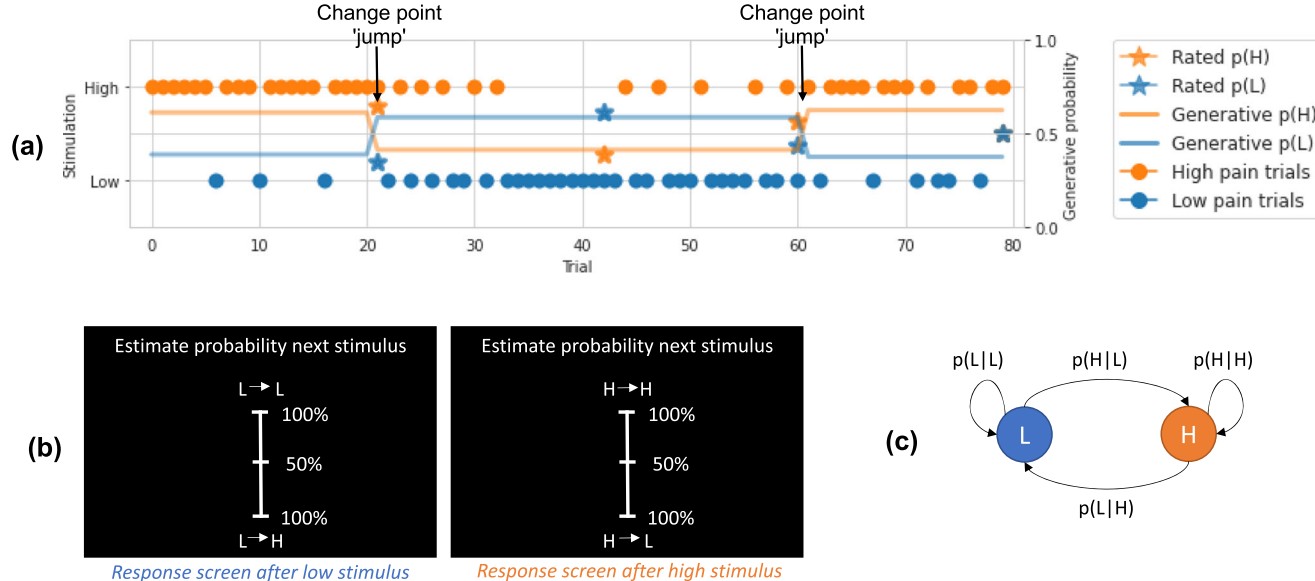

**Fig. 1 | Behavioural task and model explanation. a** Example trials from a representative participant, showing: the true probability of high (H) and low (L) stimuli given current stimuli, trial stimulation given, and participant rated probabilities. The arrows point to jump points of true probabilities, where a sudden change happens. **b** Rating screens. Occasionally, the sequence was paused and participants were asked to estimate the likelihood of the upcoming stimulus given the current one. For example, after a low stimulus participants would be asked to rate the probability of the upcoming stimulus being low (L − > L) or high (L − > H). **c** Graphical representation of the Markovian generative process of the sequence of low and high-intensity stimuli. The transition probability matrix was resampled at change points, determined by a fixed probability of a jump.

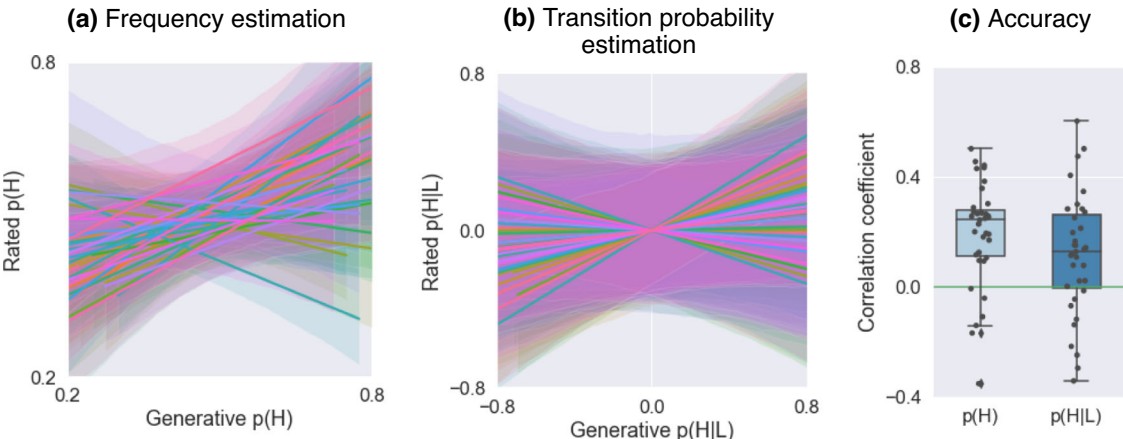

**Fig. 2 | Behavioural results. a** Relation between generative and rated frequencies, p(H), in each participant (one regression line per participant, *n* = 35). **b** Relation between generative and rated transition probabilities, p(H|L), in each participant (one regression line per participant, *n* = 35). **c** Estimation accuracy, as measured by the correlation coefficient between generative and rated probabilities: frequency p(H) and transition probability p(H|L); each circle represents one participant (*n* = 35). The boxes show the quartile of the data and the whiskers illustrate the rest of the distribution, except for points that are classified as "outliers" based on the inter-quartile range. The two sets of correlations were not significantly different, based on a two-sided z-test on Fisher-transformed correlation coefficients (z = 0.376, *p* = 0.707). Source data are provided as a Source Data file.

(frequency prediction accuracy by high pain intensity: r = −0.175, *p* = 0.337; p(H|H) prediction accuracy by high pain intensity: r = −0.178, *p* = 0.305; Supplementary Fig. 2).

### Behavioural data modelling

**Model choice.** We adopted a normative approach to identify the mathematical principles that underlie learning to predict pain sequences, based on previous evidence in other sensory domains[14,16]. We designed six computational models to address three main questions.

Firstly, we investigated whether the inference follows the rules of optimal Bayesian inference, an heuristic (a simple delta rule), or it is simply random. To this purpose, we compared a family of four Bayesian inference models[14], a basic reinforcement learning model with a fixed learning rate[17] and a baseline random model that assumes constant probabilities throughout the experiment for high and low pain respectively.

Secondly, we evaluated whether the Bayesian inference incorporates the possibility (prior probability) of sudden changes in stimulus probability or ignores such possibilities. Given that the volatility of the stimuli did not change over time, the Bayesian 'jump' models had a constant prior of the probability of a jump. The Bayesian 'fixed' models did not have any prior over the volatility of the stimuli, but had a leaky integration with an exponential decay to mimic forgetting.

Lastly, the Bayesian (jump and fixed) models differed according to the temporal statistics they inferred: the stimulus frequency or the transition probability. The Bayesian frequency models assume the sequence as generated by a Bernoulli process, where observers track how often they encountered previous stimuli. In contrast, the Bayesian transition probability models assume the sequence follows a Markov transition probability between successive stimuli, where observers estimate such transition of previous stimuli.

**Model fitting.** The selected models estimate the probability of a pain stimulus' identity in each trial. The values predicted by the model can be fitted to the subjects' probability ratings gathered during the experiment. A model is considered a good fit to the data if the total difference between the model-predicted values and the subjects' predictions is small. Within each model, free parameters were allowed to differ for individual subjects in order to minimise prediction differences. For Bayesian 'jump' models, the free parameter is the prior probability of sequence jump occurrence. For Bayesian fixed models,

the free parameters are the window length for stimuli history tracking, and an exponential decay parameter that discounts increasingly distant previous stimuli. The RL model's free parameter is the initial learning rate, and the random model assumes a fixed high pain probability that varies across subjects. The model fitting procedure minimises each subject's negative log likelihood for each model, based on residuals from a linear model that predicts subject's ratings using model predictors. The smaller the sum residual, the better fit a model's predictions are to the subject's ratings.

**Model comparison.** We compared the different models using the likelihood calculated during fitting as model evidence. Figure 3a shows model frequency, model exceedance probability, and protected exceedance probability for each model fitted (see 'Model comparison'). Both comparisons showed the winning model was the 'Bayesian jump frequency' model inferring both the frequency of pain states and their volatility, producing predictions significantly better than alternative models (Bayesian jump frequency model frequency = 0.563, exceedance probability = 0.923, protected exceedance = 0.924). Figure 3b reports the model evidence for each subject; it shows that, although the majority (*n* = 23) of participants were best fit by the model that infers the background frequency, some participants (*n* = 12) were better fit by the more sophisticated model that infers specific transition probabilities. In Supplementary Fig. 1, we show quality of the fit of the Bayesian jump frequency model for each participant.

### Neuroimaging results

We used the winning computational model to generate trial-by-trial regressors for the hemodynamic responses. The rationale behind this approach is that neural correlation of core computational components of a specific model provides evidence that and how the model is implemented in the brain[18].

First, a simple high > low pain contrast identified BOLD responses in the right thalamus, sensorimotor, premotor, supplementary motor, insula, anterior cingulate cortices and left cerebellum (with peaks in laminae V–VI), consistent with the known neuroanatomy of pain responses (Fig. 4, cluster list in Supplementary Table 2).

Next, we evaluated the neural correlates of the modelled posterior probability of high pain. For any stimulus, this reflects the newly calculated probability that the next stimulus will be high, i.e. the dynamic and probabilistic inference of high pain. Given that the

**(a)**

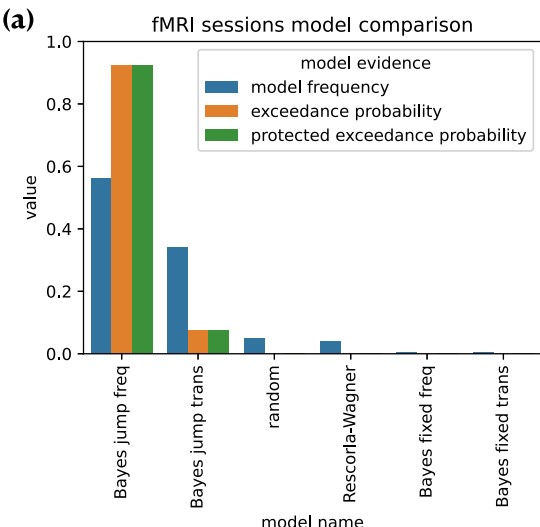

**(b)**

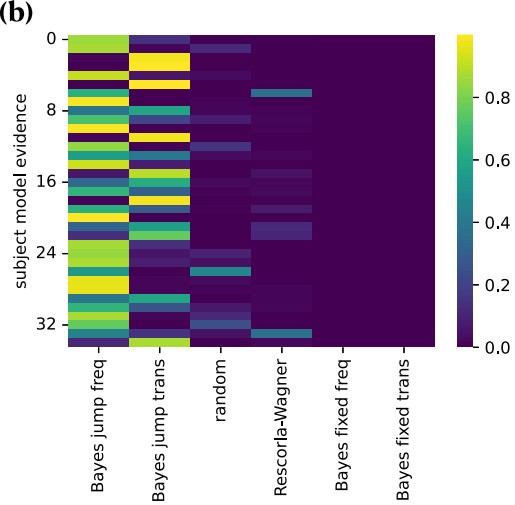

**Fig. 3 | Model comparison results. a** Bayesian model comparison based on model fitting evidence. Subjects' predictive ratings of next trial's pain intensity were fitted with posterior means from Bayesian models, values from Rescorla–Wagner (reinforcement learning) model, and random fixed probabilities. The winning model was the Bayesian jump frequency model, which assumes jumps in the sequence and infers the stimulus frequency. In our model comparison, the model frequency indicates how often a given model is used by participants; the model exceedance probability measures how likely it is that any given model is more frequent than the other models, and the protected exceedance probability is the corrected exceedance probability for observations due to chance. **b** Individual subject model evidence (each row represents a subject; colorbar indicates the model probability ranging from 0 to 1). Source data are provided as a Source Data file.

## Encoding of stimulus intensity

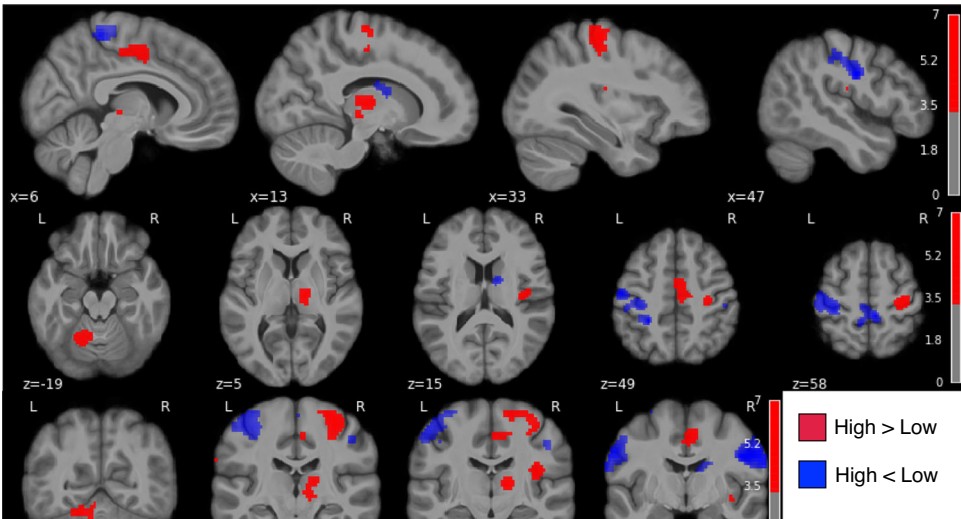

**Fig. 4 | Brain responses to noxious stimuli.** Red: high > low pain stimuli, blue: high < low pain stimuli. Two-sided statistics corrected for the false discovery rate (FDR) at level $p < 0.001$; colorbar shows Z scores > 3.3.

predicted probabilities of high and low pain are reciprocal, their neural correlates can be revealed by using positive and negative contrasts. The prediction of high pain frequency was associated with BOLD responses in the bilateral primary and secondary somatosensory cortex, primary motor cortex, caudate and putamen (pink clusters in Fig. 5, Table 1). The prediction of low pain frequency implicated the right (controlateral to stimulation) sensorimotor cortex, the supplementary motor cortex, dorsal anterior cingulate cortex, thalamus and posterior insular-opercular cortex bilaterally (green clusters in Fig. 5, Table 1).

Uncertainty signals, quantified by the variability (SD) of the posterior probability distribution of high pain, were found in a right superior parietal region, bordering with the supramarginal gyrus (Fig. 6a and Table 2). The negative contrast of the posterior SD did not yield any significant cluster.

A key aspect of the Bayesian model is that it provides a metric of the model update, quantified as the Kullback–Leibler (KL) divergence between successive trials' posterior distribution. The KL divergence increases when the two successive posteriors are more different from each other, and decreases when the posteriors are similar. We found that the KL divergence was associated with BOLD responses in left premotor cortex, bilateral dorsolateral prefrontal cortex, superior parietal lobe, supramarginal gyrus, and left somatosensory cortex (Fig. 6b, Table 2). For completeness, we report the negative contrast in Supplementary Fig. 3 and Supplementary Table 3. Figure 7 overlays the posterior probability of pain with its uncertainty and update (KL

**Temporal statistical inference of pain intensity**

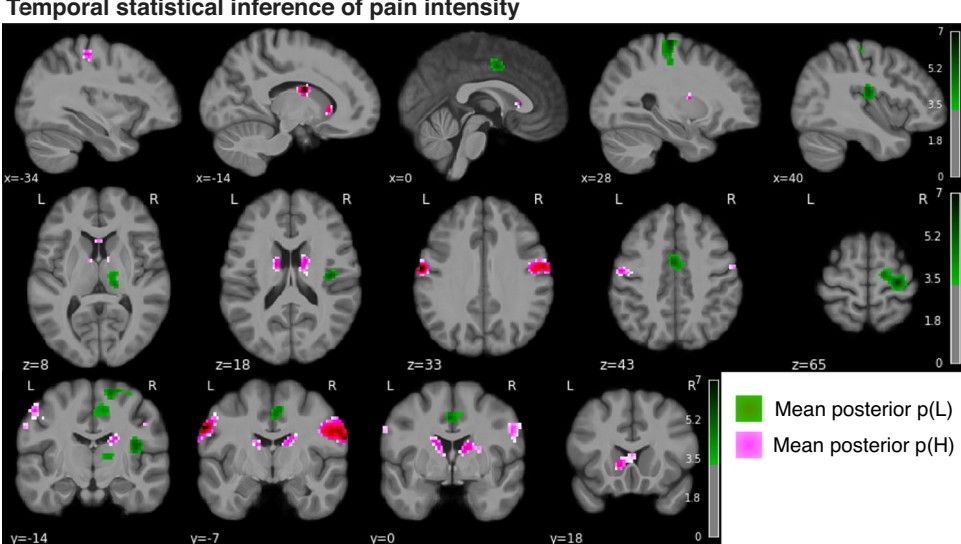

**Fig. 5 | Brain activity associated with the temporal statistical inference of pain intensity.** Neural correlates of the mean posterior probability of low pain (green) and high pain (pink) in the Bayesian jump frequency model (two-sided statistics, FDR corrected $p < 0.001$, colorbar shows Z scores $> 3.3$).

divergence). This shows that the temporal prediction of high pain and its update activate distinct, although neighbouring regions in the sensorimotor and premotor cortex, bilaterally. In contrast, the uncertainty of pain predictions activates a right superior parietal region that partially overlaps with the neural correlates of model update.

## Discussion

Pain is typically uncertain, and this is most often true when pain persists after an injury. When this happens, the brain needs to be able to track changes in intensity and patterns over time, in order to predict what will happen next and decide what to do about it. Here we show that the human brain can generate explicit predictions about the likelihood of forthcoming pain, in absence of external cues. Pain predictions are best described by an optimal Bayesian inference model with dynamic update of beliefs, allowing explicit prediction of the probability of forthcoming pain at any moment in time. Using neuroimaging, we found distinct neural correlates for the probabilistic, predictive inference of pain and its update. Predictions (i.e. mean posterior probability) of high pain intensity are encoded in the bilateral, primary somatosensory and motor regions, secondary somatosensory cortex, caudate and putamen, whereas predictions of low pain intensity involve the controlateral sensorimotor cortex, the supplementary motor cortex, dorsal anterior cingulate cortex, bilateral thalamus and posterior insular-opercular cortex. The signal representing the update of the probabilistic model localises in adjacent premotor and superior parietal cortex. The superior parietal cortex is also implicated in the computation of the uncertainty of the probabilistic inference of pain. Overall, the results show that cortical regions typically associated with the sensory processing of pain (primary and secondary somatosensory cortices) encode how likely different pain intensities are to occur at any moment in time, in the absence of any other cues or information; the uncertainty of this inference is encoded in the superior parietal cortex and used by a network of parietal-prefrontal regions to update the temporal statistical representation of pain intensity.

The ability of the brain to extract regularities from temporal sequences is well-documented in other sensory domains such as vision and audition[8,11]. However, pain is a fundamentally different system with intrinsic motivational value and direct impact on the state of the body[19–21]. Pain can constrain cognitive functions, such as working

memory and attention[22–24]. Furthermore, the cortical representation of pain is distributed[25] and a primary brain region for pain has not been found[26].

We show that pain predictions are best described by an optimal Bayesian inference model, tracking the frequency of pain states and their volatility based on past experience. A more complex strategy involves inferring higher level statistical patterns within these sequences, i.e. representing all the transition probabilities between different states. It has been shown that optimal inference of transition probabilities can be achieved using similar paradigms with visual and auditory stimuli[14]. Although this model fits 1/3 of subjects best, overall it was not favoured over the simpler frequency learning model, which best describes the behaviour of ~2/3 of our sample (Fig. 3). At this stage it is not clear whether this is because of stable inter-individual differences, or whether given more time, more participants would be able to learn specific transition probabilities. However, it is worth noting that stable, individual differences in learning strategy have been previously reported in visual statistical learning[13,27]. In supplementary analyses, we show that the neural correlates of both frequency and transition probability learning were generally comparable between the subgroups of participants who favoured a frequency inference strategy and those who preferred a transition probability strategy (Supplementary Notes 6–7).

As in other domains, we focused on conscious judgement of the relative overall probability of pain, as opposed to looking at autonomic responses or other physiological measures of pain prediction; this was done to allow direct comparisons between the participant's predictions and ideal observers[14]. We did not test whether statistical learning for pain is automatic but, based on previous work, we expect it to happen spontaneously, without requiring the need to explicitly report stimulus probabilities. Indeed, statistical learning of visual or auditory inputs has been reported in songbirds, primates and newborns[28–32]. Furthermore, previous studies have shown implicit expectation effects from sequences of stimuli in humans[33–35].

The present evidence in support of Bayesian inference is broadly consistent with previous work on the learning of a cognitive model or acquisition of explicit contingency knowledge across modalities, including pain[36–38]. This reflects a fundamentally different process to pain response learning—either in Pavlovian conditioning where simple autonomic, physiological or motoric responses are acquired, or basic stimulus-response (instrumental / operant) avoidance or

**Table 1 | Activation clusters associated with the mean posterior p(High pain) and p(Low pain) of the Bayesian jump frequency model**

|  | Cluster ID | X | Y | Z | Peak stat | Cluster size (mm³) |
|---|---|---|---|---|---|---|
| p(H) |  |  |  |  |  |  |
| 0 | 1 | 66 | −7 | 27 | 6.477 | 4402 |
| 1 | 1a | 52 | −7 | 33 | 5.787 |  |
| 2 | 2 | −62 | −7 | 33 | 5.924 | 2408 |
| 3 | 2a | −46 | −12 | 43 | 4.002 |  |
| 4 | 3 | 21 | −12 | 24 | 4.885 | 1491 |
| 5 | 3a | 11 | −2 | 15 | 4.197 |  |
| 6 | 3b | 13 | −7 | 21 | 4.140 |  |
| p(L) |  |  |  |  |  |  |
| 0 | 1 | 28 | −22 | 65 | 6.158 | 5265 |
| 1 | 1a | 30 | −19 | 49 | 6.012 |  |
| 2 | 1b | 16 | −12 | 65 | 5.148 |  |
| 3 | 1c | 23 | −10 | 68 | 4.575 |  |
| 4 | 2 | 37 | −17 | 15 | 6.013 | 1311 |
| 5 | 3 | 13 | −22 | 8 | 5.407 | 1886 |
| 6 | 4 | 11 | −17 | 49 | 4.683 | 2372 |
| 7 | 4a | 0 | −2 | 43 | 4.439 |  |

**Table 2 | Activation clusters positively associated with the uncertainty (SD posterior) and update (KL divergence) of the Bayesian jump frequency model**

|  | Cluster ID | X | Y | Z | Peak stat | Cluster size (mm³) |
|---|---|---|---|---|---|---|
| Uncertainty |  |  |  |  |  |  |
| 0 | 1 | 40 | −48 | 58 | 4.311 | 1186 |
| 1 | 1a | 47 | −38 | 58 | 4.168 |  |
| 2 | 1b | 33 | −41 | 43 | 3.745 |  |
| 3 | 2 | 28 | −58 | 49 | 4.084 | 736 |
| Update |  |  |  |  |  |  |
| 0 | 1 | −58 | 6 | 36 | 6.191 | 11034 |
| 1 | 1a | −26 | −2 | 49 | 5.945 |  |
| 2 | 1b | −60 | 4 | 21 | 4.935 |  |
| 3 | 1c | −43 | 0 | 55 | 4.516 |  |
| 4 | 2 | −46 | −41 | 40 | 6.098 | 5193 |
| 5 | 2a | −36 | −50 | 52 | 5.438 |  |
| 6 | 2b | −50 | −41 | 55 | 3.789 |  |
| 7 | 3 | 59 | 11 | 24 | 5.308 | 1886 |
| 8 | 4 | 47 | −41 | 58 | 5.295 | 6128 |
| 9 | 4a | 37 | −50 | 52 | 4.972 |  |
| 10 | 4b | 37 | −58 | 61 | 4.460 |  |
| 11 | 4c | 30 | −65 | 61 | 4.255 |  |
| 12 | 5 | −62 | −17 | 33 | 4.814 | 1797 |
| 13 | 5a | −50 | −24 | 33 | 4.584 |  |
| 14 | 5b | −46 | −29 | 27 | 3.849 |  |

escape response learning. These behaviours are usually best captured by reinforcement learning models such as temporal difference learning[21], and reflect a computationally different process[39]. Having said that, such error-driven learning models have been applied to statistical learning paradigms in other domains before[40], and so here we were able to directly demonstrate that they provided a worse fit than Bayesian inference models (Fig. 3). In contrast to simple reinforcement learning models, Bayesian models allow to build an internal, hierarchical representation of the temporal statistics of the environment that can support a range of cognitive functions[14,41,42].

A key benefit of the computational approach is that it allows us to accurately map underlying operations of pain information processing to their neural substrates[43]. Our study shows that the probabilistic inference of pain frequency is encoded in somatosensory processing regions, such as the primary somatosensory cortex and posterior insula/operculum; it also involves supramodal regions, such as premotor cortex and dorsal striatum (Fig. 5). This is broadly consistent with the view that statistical learning involves both sensory and supramodal regions[9].

A specific facet of the Bayesian model is the representation of an uncertainty signal, i.e. the posterior SD, and a model update signal, defined as the statistical KL divergence between consecutive posterior distributions. This captures the extent to which a model is updated when an incoming pain signal deviates from that expected, taking into account the uncertainty inherent in the original prediction. In our task, the uncertainty of the prediction was encoded in a right superior parietal region, which partially overlapped with a wider parietal region associated with the encoding of the model update (Figs. 6, 7). This emphasises the close relationship between uncertainty and learning in Bayesian inference[44]. A previous study on statistical learning in other sensory domains reported that a more posterior, intraparietal region was associated with the precision of the temporal inference[15]. The role of the superior parietal cortex in uncertainty representation is also evident in other memory-based decision-making tasks; e.g. the superior parietal cortex was found to be more active for low vs. high confidence judgements[45–47]. In addition to the parietal cortex, the model update signal was encoded in the left premotor cortex and bilateral dorsolateral prefrontal cortex

(Fig. 6), which neighboured regions activated by pain predictions (Fig. 7). This is particularly interesting, as the premotor cortex sits along a hierarchy of reciprocally and highly interconnected regions within the sensorimotor cortex. The premotor cortex has also been implicated in the computation of an update signal in visual and auditory statistical learning tasks[15].

In conclusion, our study demonstrates that the nociceptive system generates probabilistic predictions about the background temporal statistics of pain states, in absence of external cues, and this is best described by a Bayesian inference strategy. This extends both current anatomical and functional concepts of what is conventionally considered a 'sensory pain pathway', to include the encoding not just of stimulus intensity[48,49] and location[50], but also the generation of dynamic internal models of the temporal statistics of pain intensity levels. Future studies will need to determine whether temporal statistical predictions modulate pain perception, similarly to other kinds of pain expectations[2,51,52]. More broadly, temporal statistical learning is likely to be most important after injury, when continuous streams of fluctuating signals ascend nociceptive afferents to the brain, and their underlying pattern may hold important clues as to the nature of the injury, its future evolution, and its broader semantic meaning in terms of the survival and prospects of the individual. It is therefore possible that the underlying computational process might go awry in certain instances of chronic pain, especially when instrumental actions can be performed that might influence the pattern of pain intensity[37,53]. Thus, future studies could explore both how temporal statistical learning interacts with pain perception and controllability, as well as its application to clinical pain.

## Methods
### Participants
Thirty-five healthy participants (17 females; mean age 27.4 years old; age range 18–45 years) took part in two experimental sessions, 2–3 days apart: a pain-tuning and training session and an MRI session.

## (a) Uncertainty of the inference (SD posterior)

## (b) Jump frequency model update

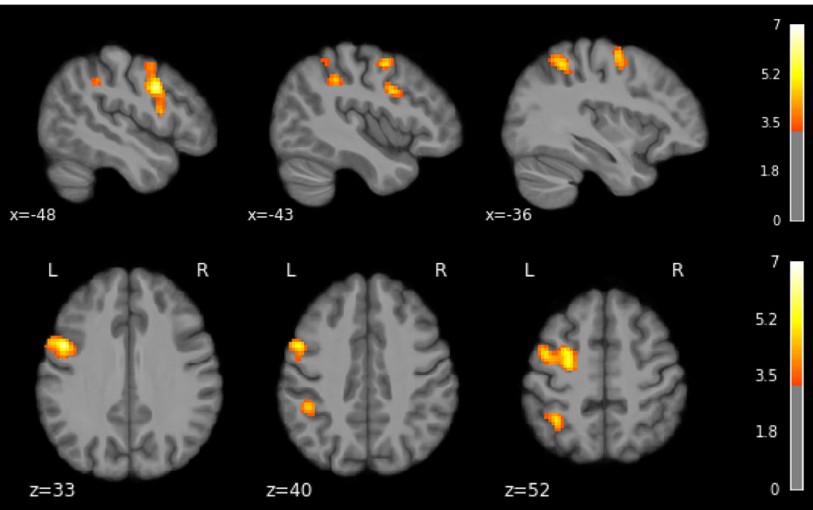

**Fig. 6 | Uncertainty and learning signals. a** Uncertainty (SD) of the posterior probability of high pain in the Bayesian jump frequency model was associated with activations in the right superior parietal cortex (FDR corrected $p < 0.001$, colorbar shows Z scores > 3.3). **b** Neural activity associated with the model update, i.e. the Kullback–Leibler (KL) divergence between posteriors from successive trials (positive contrast, FDR corrected $p < 0.001$, colorbar shows Z scores > 3.3).

Each participant gave informed consent according to procedures approved by University of Cambridge ethics committee (PRE.2018.046).

## Protocol

The electrical stimuli were generated using a DS5 isolated bipolar current stimulator (Digitimer), delivered to surface electrodes placed on the index and middle fingers of the left hand. All participants underwent a standardised intensity work-up procedure at the start of each testing day, in order to match subjective pain levels across sessions to a low-intensity level (just above pain detection threshold) and a high-intensity level that was reported to be painful but bearable (>4 out of 10 on a VAS ranging from 0 ['no pain'] to 10 ['worst imaginable pain']). The stimulus delivery setup was identical for lab-based and MR sessions. After identifying appropriate intensity levels, we checked that discrimination accuracy was >95% in a short sequence of 20 randomised stimuli. This was done to ensure that uncertainty in the sequence task would derive from the temporal order of the stimuli rather than their current intensity level or discriminability. If needed, we tweaked the stimulus intensities to achieve our target discriminability. Next, we gave the task instructions to each participants (openly available[54]).

After receiving a shock on trial $t$, subjects were asked to predict the probability of receiving a stimulus of the same or different intensity on the upcoming trial (trial $t + 1$). We informed participants that in the task they "would receive two kinds of stimuli, a low-intensity shock and a high-intensity shock. The L and H stimuli would be presented in a sequence, in an order set by the computer. After each stimulus, the following stimulus intensity could be either the

same or change. The computer sets the probability that after a given stimulus (for example L) there would be either L or H" (we showed a visual representation of this example). We asked participants to "always try to guess the probability that after each stimulus there will be the same or a different one" and we informed them that "the computer sometimes changes its settings and sets new probabilities", so to pay attention all the time. We also told them the sequence would be paused occasionally in order to collect probability estimates from participants using the scale depicted in Fig. 1. A white fixation cross was displayed on a dark screen throughout the trial, except when a response was requested every 12–18 trials. The interstimulus interval was 2.8–3 seconds. There were 300 stimuli in each block, lasting ~8 min. Average intensity ratings for each stimulus level were collected after each block during a short break. Low-intensity stimuli were felt by participants as barely painful, rated on average 1.39 (SD 0.77) on a scale ranging from 0 (no pain) to 10 (worst pain imaginable). In contrast, high-intensity stimuli were rated as more than 4 times higher than low-intensity stimuli (mean 5.74, SD 4.85). Participants were given 4 blocks of practice, 2–3 days prior the imaging sessions and 5 blocks (1500 stimuli) during task fMRI.

A unique sequence was generated every time the experiment was launched as in ref. 14. L and H stimuli were drawn randomly from a 2 × 2 transition probability matrix, which remained constant for a number of trials (chunks). The probability of a change was 0.014. Chunks had to be >5 and <200 trials long. In each chunk, transition probabilities were sampled independently and uniformly in the 0.15–0.85 range (in steps of 0.05), with the constraint that at least one of the two transition probabilities must be >/< 0.2 than in the previous chunk. Participants

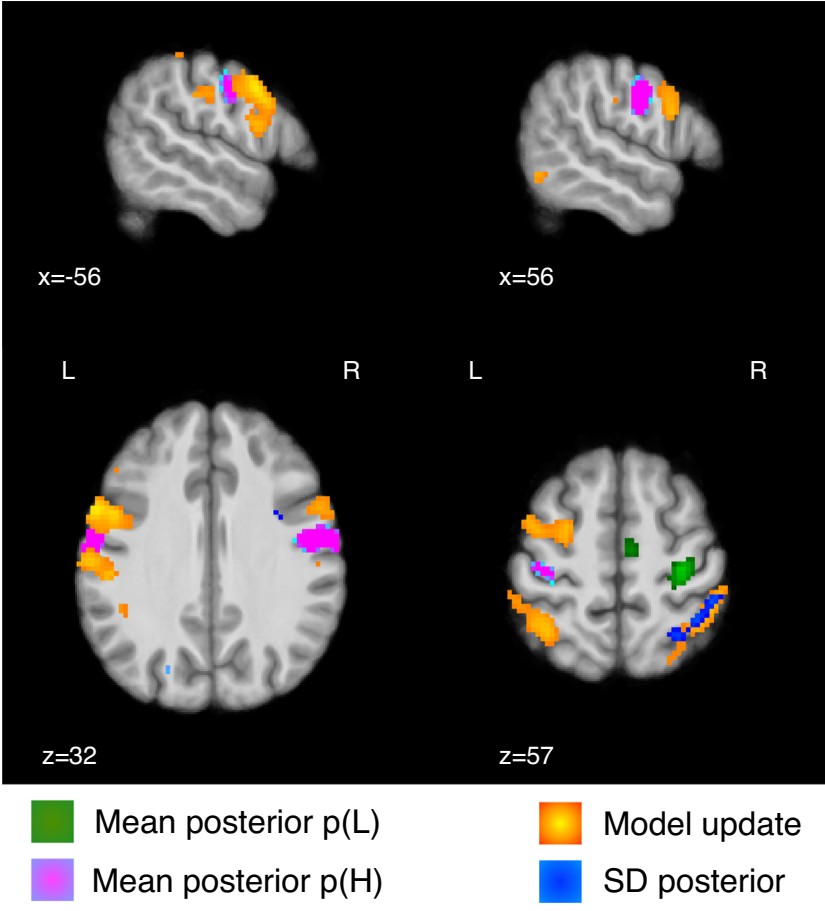

**Fig. 7 | Statistical inference and model update activate adjacent sensorimotor and premotor regions.** Overlay of the temporal prediction (mean posterior probability) of low (green) and high pain (pink), their uncertainty (SD posterior probability, blue) and the model update (KL divergence between successive posterior distributions, red-yellow); FDR corrected $p < 0.001$, colorbar shows Z scores > 3.3.

were not informed when the matrix was resampled, and a new chunk started.

At the end of the task we asked participants for general comments about the strategy they used. A minority of participants reported to have no clue about the stimulus sequence and were rather bored by the task. The majority of subjects thought that it was difficult to predict the sequence but they were starting to get a sense of the temporal pattern after a while. Some tried to use strategies like counting, but they tended to abandon it to favour a more spontaneous approach of feeling the "flow" or the "rhythm" of the sequence.

Behavioural data analysis were conducted with Python packages pandas (pypi version 1.1.3) and scipy (pypi version 1.5.3). Effect size was calculated as Cohen's d for *t*-tests.

## Computational modelling of temporal statistical learning

**Learning models.** The models used in comparison are listed as followed:

### Random (baseline model)

Probabilities are assumed fixed and reciprocal for high and low stimuli

$$p_h = 1 - p_l \tag{1}$$

where $p_l$ was fitted as a free parameter. Uncertainty was assumed to be fixed.

### Rescorla–Wagner (RW model)

Rated probabilities are assumed to be state values, which were updated as

$$V_{t+1} \leftarrow V_t + \alpha(R_t - V_t) \tag{2}$$

where $R_t = 1$ if stimulus was low, and 0 otherwise. The learning rate $\alpha$ was fitted as free parameter[17].

### Bayesian models

Bayesian models update each trial with stimulus identity information to obtain upcoming trial probability from the posterior distribution[14]. Using Bayes' rule, the model parameter $\theta_t$ is estimated at each trial $t$ provided previous observations $y_{1:t}$ (sequence of high or low pain), given a model $M$.

$$p(\theta_t|y_{1:t},M) \sim p(y_{1:t}|\theta_t,M)p(\theta_t,M) \tag{3}$$

Stimulus information can either be the frequency or transition of the binary sequence. There are 'fixed' models that assume no sudden jump in stimuli probabilities, and 'jump' models that assume the opposite. The four combinations were fitted and compared.

### Fixed frequency model

For fixed models, the likelihood of parameters $\theta$ follows a Beta distribution with parameters $N_h + 1$ and $N_l + 1$, where $N_h$ and $N_l$ are the numbers of high and low pain in the sequence $y_{1:t}$. Given that the prior is also a flat Beta distribution with parameters [1,1], the posterior can be analytically obtained with:

$$p(\theta|y_{1:t}) = \text{Beta}(\theta|N_h + 1, N_l + 1) \tag{4}$$

The likelihood of a sequence $y_{1:t}$ given model parameters $\theta$ can be calculated as:

$$p(y_{1:t}|\theta) = p(y_1|\theta) \prod_{i=2}^{t} p(y_i|\theta, y_{i-1}) \tag{5}$$

Finally, the posterior probability of a stimulus occurring in the next trial can be estimated with Bayes' rule:

$$p(y_{t+1}|y_{1:t}) = \int p(y_{t+1}|\theta, y_t) p(\theta|y_{1:t}) d\theta \tag{6}$$

Priors "window" and "decay" were fitted as free parameters. "Window" is the previous $n$ trials where the frequency of stimuli was estimated, and "decay" is the previous $n$ trials where the frequency of stimuli further from current trial was discounted following an exponential decay.

When "window" $w$ is applied, then $N_h$ and $N_l$ are counted within the window of $w$ trials $y_{t-w,t}$. When "decay" $d$ is applied, an exponential decay factor $e^{(-\frac{k}{d})}$ is applied to the $k$ trials before their sum is calculated. Both "window" and "decay" were used simultaneously.

### Fixed transition model

Priors "window" and "decay" were fitted as free parameters as in the Fixed frequency model above, however, the transition probability was estimated instead of the frequency. The likelihood of a stimulus now depends on the estimated transition probability vector $\theta \sim [\theta_{h|l}, \theta_{l|h}]$ and the previous stimulus pairs $N \sim [N_{h|l}, N_{h|h}]$. Given that both likelihood and prior can be represented using Beta distributions as before, the posterior result can be analytically obtained as:

$$p(\theta|y_{1:t}) = \text{Beta}(\theta_{h|l}|N_{h|l}+1, N_{l|l}+1)\text{Beta}(\theta_{l|h}|N_{l|h}+1, N_{h|h}+1) \tag{7}$$

### Jump frequency model

In jump models, parameter $\theta$ is no longer fixed, instead it can change from one trial to another with a probability of $p_{jump}$. Prior $p_{jump}$ was fitted as a free parameter, representing the subject's assumed probability of a jump occurring during the sequence of stimuli (e.g. a high $p_{jump}$ assumes the sequence can reverse quickly from a low pain majority to a high pain majority). The model can be approximated as a Hidden Markov Model (HMM) in order to compute the joint distribution of $\theta$ and observed stimuli iteratively,

$$p(\theta_{t+1}, y_{1:t+1}) = p(y_{t+1}|\theta_{t+1}, y_t) \int p(\theta_t, y_{1:t}) p(\theta_{t+1}|\theta_t) d\theta_t \tag{8}$$

where the integral term captures the change in $\theta$ from one observation $t$ to the next $t+1$, with probability $(1-p_{jump})$ of staying the same and probability $p_{jump}$ of changing. This integral can be calculated numerically within a discretised grid. The posterior probability of a stimulus occurring in the next trial can then be calculated using Bayes' rule as

$$\begin{aligned} p(y_{t+1}|y_{1:t}) &= \int p(y_{t+1}|\theta_{t+1}) p(\theta_{t+1}|y_{1:t}\theta_{t+1}) \\ &= \int p(y_{t+1}|\theta_{t+1}) \left[ \int p(\theta_t|y_{1:t}) p(\theta_{t+1}|\theta_t) d\theta_t \right] d\theta_{t+1} \\ &= \int p(y_{t+1}|\theta_{t+1}) \left[ (1-p_{jump}) p(\theta_{t+1}=\theta_t|y_{1:t}) + p_{jump} p(\theta_0) \right] d\theta_{t+1} \end{aligned} \tag{9}$$

### Jump transition model

Similarly to the jump frequency model above, prior $p_{jump}$ was fitted as a free parameter, but estimating transition probabilities instead of frequency. The difference is the stimulus at trial $y_{t+1}$ now dependent on the stimulus at the previous trial, hence the addition of the term $y_t$ in the joint distribution term, shown below.

$$p(y_{t+1}|y_{1:t}) = \int p(y_{t+1}|\theta_{t+1}, y_t) \left[ (1-p_{jump}) p(\theta_{t+1}=\theta_t|y_{1:t}) + p_{jump} p(\theta_0) \right] d\theta_{t+1} \tag{10}$$

### KL divergence.

Kullback−Leibler (KL) divergence quantifies the distance between two probability distributions. In the current context, it measures the difference between the posterior probability distributions of successive trials. It is calculated as

$$D_{KL}(P \parallel Q) = \sum_{x \in \mathcal{X}} P(x) \log\left(\frac{P(x)}{Q(x)}\right) \tag{11}$$

where $P$ and $Q$ represent the two discrete posterior probability distributions calculated in discretised grids $\mathcal{X}$. KL divergence can be used to represent information gains when updating after successive trials[15].

### Subject rated probability.

For each individual subject, model-predicted probabilities $p_k$ from the trial $k$ were used as predictors in the regression:

$$y_k \sim \beta_0 + \beta_1 \cdot p_k(M_i, \theta_i) + \beta_2 \cdot N_s + \epsilon \tag{12}$$

where $y_k$ is the subject rated probabilities, $M_i$ is the $i$th candidate model, $N_s$ is the session number within subject, $\beta_0$, $\beta_1$, $\beta_2$ and $\theta_i$ are free parameters to be fitted and $\epsilon$ is normally distributed noise added to avoid fitting errors[55].

### Model fitting.

To estimate the model-free parameters from data, Bayesian information criteria (BIC) values were calculated as:

$$\text{BIC} = n \cdot \log \hat{\sigma}_\epsilon^2 + k \cdot \log n \tag{13}$$

$$\hat{\sigma}_\epsilon^2 = \min \frac{1}{n} \sum_{k-1}^{n} (y_k - \hat{y}_k) \tag{14}$$

where $\hat{\sigma}^2$ is the squared residual from the linear model above that relates subject ratings to model-predicted probabilities, and $n$ is the number of free parameters fitted.

We use *fmincon* in MATLAB to minimise the BIC (as approximate for negative log likelihood[55]) for each subject/model. The procedure was repeated 100 times with different parameter initialisation, and the mean results of these repetitions were taken as the fitted parameters and minimised log likelihoods. Model's parameter recovery is reported in Supplementary Note 1.

### Model comparison.

In general, the best fit model was defined as the candidate model with the lowest averaged BIC. We conducted a random effect analysis with the VBA toolbox[56], where fitted log likelihoods from each subject/model pair were used as model evidence. With this approach, model evidence was treated as random effects that could differ between individuals. This comparison produces model frequency (how often a given model is used by individuals), model exceedance probability (how likely it is that any given model is more frequent than all other models in the comparison set), and protected exceedance probability (corrected exceedance probability for observations due to chance)[57,58]. These values are correlated and would be considered together when selecting the best fit model.

## Neuroimaging data

**Data acquisition.** First, we collected a T1-weighted MPRAGE structural scan (voxel size 1 mm isotropic) on a 3T Siemens Magnetom Skyra (Siemens Healthcare), equipped with a 32-channel head coil (Wolfson Brain Imaging Centre, Cambridge). Then we collected 5 task fMRI sessions of 246 volumes using a gradient echo planar imaging (EPI) sequence (TR = 2000 ms, TE = 23 ms, flip angle = 78°, slices per volume = 31, Grappa 2, voxel size 2.4 mm isotropic, A > P phase-encoding; this included four dummy volumes, in addition to those pre-

discarded by the scanner). In order to correct for inhomogeneities in the static magnetic field, we imaged 4 volumes using an EPI sequence identical to that used in task fMRI, inverted in the posterior-to-anterior phase-encoding direction. Full sequence metadata are available[59].

**Preprocessing.** Imaging data were preprocessed using fmriprep (pypi version: 20.1.1, RRID:SCR_016216) with Freesurfer option disabled, within its Docker container. Processed functional images had first four dummy scans removed, and then smoothed with an 8 mm Gaussian filter in SPM12.

**Generalised linear model analysis.** Nipype (pypi version: 1.5.1) was used for all fMRI processing and analysis within its published Docker container. Nipype is a python package that wraps around fMRI analysis tools including SPM12 and FLS in a Debian environment.

First and second level GLM analyses were conducted using SPM12 through nipype. In all first level analyses, 25 regressors of no interest were included from fmriprep confounds output: CSF, white matter, global signal, dvars, std_dvars, framewise displacement, rmsd, 6 a_comp_cor with corresponding cosine components, translation in 3 axis and rotation in 3 axis. Sessions within subject are not concatenated.

In second level analyses, all first level contrasts were entered into a one-sample $t$-test, with group subject mask applied. The default FDR threshold used was 0.001 (set in Nipype threshold node height_threshold = 0.001).

For visualisation and cluster statistics extraction, nilearn (pypi version: 1.6.1) was used. A cluster extent of 10 voxels was applied. Visualised slice coordinates were chosen based on identified cluster peaks. Activation clusters were overlaid on top of a subject averaged anatomical scan normalised to MNI152 space as output by fmriprep.

**GLM design.** All imaging results were obtained from a single GLM model. We investigated neural correlates using the winning Bayesian jump frequency model. All model predictors were generated with the group mean fitted parameters in order to minimise noise. First level regressors include the onset times for all trials, high pain trials and low pain trials (duration = 0). The all trial regressor was parametrically modulated by model-predicted posterior mean of high pain, the KL divergence between successive posterior distributions on jump probability, and the posterior SD of high pain.

For second level analysis, both positive and negative T-contrasts were obtained for posterior mean, KL divergence and uncertainty parametric modulators, across all the first level contrast images from all subjects. A group mean brain mask was applied to exclude activations outside the brain. Given that high and low pain are reciprocal in probabilities, a negative contrast of posterior mean of low pain would be equivalent to the posterior mean of high pain. In addition, high and low pain comparisons were done using a subtracting T-contrast between high and low pain trial regressors. We corrected for multiple comparisons with a cluster-wise FDR threshold of $p < 0.001$ for both parametric modulator analyses, reporting only clusters that survived this correction.

**Reporting summary**
Further information on research design is available in the Nature Research Reporting Summary linked to this article.

## Data availability
Raw functional imaging data are deposited at https://doi.org/10.18112/openneuro.ds003836.v1.0.0[59] and derived statistical maps are available at https://neurovault.org/collections/12827/. Sequence generation, task instructions and behavioural data are openly available https://zenodo.org/record/6997897. Source data are provided with this paper.

## Code availability
Analysis code is openly available at https://zenodo.org/record/6997887[54].

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

## Acknowledgements

F.M. was funded by a Medical Research Council Career Development Award (MR/T010614/1). B.S. was funded by by Wellcome (214251/Z/18/Z), Versus Arthritis (21537) and IITP (MSIT 2019-0-01371). We are grateful to Professor Máté Lengyel, Professor Zoe Kourtzi, Dr Michael Lee and Dr Dounia Mulders for helpful discussions about the study, and to the staff of the Wolfson Brain Imaging Centre for their support during data collection. For the purpose of open access, the author has applied a Creative Commons Attribution (CC BY) licence to any Author Accepted Manuscript version arising from this submission.

## Author contributions

F.M. and B.S. designed the study. F.M. collected the data and S.Z. analysed the data. All authors wrote the paper.

## Competing interests

The authors declare no competing interests.
