## [Peer Review File · Nature Communications]

Computational and neural mechanisms of statistical pain learningREVIEWER COMMENTS

Reviewer #1 (Remarks to the Author):

This manuscript examined the computational and neural mechanisms of temporal statistical learning in the pain experience. The authors showed that participants used Bayesian inference with the dynamic update of beliefs to predict the intensity of upcoming pain, in the absence of any external cues. The different components of the Bayesian modeling, including overall posterior probability (mean), its uncertainty (standard deviation), and model update (consecutive KL divergence) are distinctively related to the activations of classical pain-processing brain regions and additional fronto-parietal and pre-motor regions. Overall, this work is very clear and interesting, and provides novel information to pain neuroimaging researchers. I have a few comments about this manuscript, on which I elaborate below.

1. I'm not fully convinced about the rationale for the frequency modeling. In this study, participants were asked to guess the conditional probability of upcoming pain intensity (i.e., $P(H|L)$ or $P(L|H)$) always. If the participants followed the instruction thoroughly, it is likely that they always tried to infer the underlying transition probability, not the frequency statistics (i.e., $P(H)$ or $P(L)$). The estimation of transition probability also can help prediction because the stimuli sequences were generated based on the dynamically changing transition probability itself. However, the authors set the frequency model as their main hypothesis, and the transition model as a supplementary one. I'm not sure this is the most principled way, considering the experimental design of this study. If the authors expected that some participants mainly use frequency statistics for prediction of pain intensity despite the previous task instruction to estimate transition probability, I think at least there should be some questionnaires to ask participants which strategy they primarily used for prediction in the scanner.

2. In line with 1, the selected model was based on frequency statistics, but the behavioral results were about the transitional probability. This seems inconsistent to me. Also, the overall correlation between subjective rating and true transitional probability is mediocre, and there were even negative correlations for some participants. Would the Bayesian jump transition model be selected, if the overall correlation was very high? I'm wondering whether the authors can provide some clarification on this.

3. I'm wondering whether the predicted transitional probability from the Bayesian jump frequency model actually resembles the subjective rating of transitional probability, like Figure 4 of Meyniel et al., 2015, Plos Comp Biol.

4. I think the authors may want to correct some typos.

1) Line 68. Figure 2a -> Figure 1b

2) Lines 86, 89, 93, 94. Figure 2 -> Figure 1

Reviewer #2 (Remarks to the Author):

Overall this is a well conducted, methodologically interesting paper examining explicit predictions of upcoming pain. It isn't clear to me, however, how important this is to our understanding of how pain is processed. Is it actually common for people to consciously process pain probabilities or is this more likely to be an implicit process? Isn't it possible that the cognitive evaluations performed here aren't specific to the pain system and that similar processes would occur attempting to evaluate probabilities of any two different stimuli? While the activations within pain regions might argue against this, we also don't have any indications that the processes being examined have any effect on the processing of the stimuli, so the relevance to our understanding of pain processing is unclear.

-In the discussion it is stated that "we show that temporal inferences of pain are generated using optimal Bayesian inference". As a minor point, this phrasing confuses a model that approximates how the brain works with how the brain actually works. In a larger sense, can we really draw conclusions about how well the model approximates what happens? The model evaluation statistics seem relative – the chosen model outperforms other models tested, but this doesn't necessarily indicate that it is a good fit overall.

-It isn't clear why the authors predict that encoding of temporal probabilities should be encoded in pain processing regions.

-In Figure 2, prediction success is shown for an "example participant" but this participant is clearly not a typical example of prediction success but, rather, one of the most successful (their correlation is nearly 40 points higher than the mean). This gives a false impression of performance on this task.

Reviewer #3 (Remarks to the Author):

The study by Mancini and colleagues used a series of electrical stimuli to investigate the prediction of the intensity of painful stimuli in healthy subjects. Most of the participants were able to learn the likelihood of high or low-intensity stimulation. The authors used different modelling approaches to study the underlying cortical processes of this prediction. The general approach has been tested before for the visual domain by a different group.

Therefore, the entire approach has been validated already and the other methods appear sound. The prediction of pain has some behavioural relevance and the investigation can be of utmost significance - not only for the pain researchers but for the entire neuroscience community. I particularly like the idea that the predictions are NOT based on cues; this approximates real-life conditions, where visual cues (red squares or blue triangles etc.) rarely indicate the intensity of upcoming pain. I very much

acknowledge the availability of data and code. The results support all conclusions and claims. The work can be reproduced through the provided data and code and meets the standards that can be expected in neuroimaging.

In my view, the manuscript is acceptable as it is. However, a few points could be clarified in order to rule out potential questions from some future readers. This could be done in this or in a subsequent publication.

Suggestions:

There are some inconsistencies regarding the statistical threshold (FDR, FWE). I would suggest using the same method throughout the manuscript.

The prediction accuracy varies substantially across participants and there are apparently two different groups of participants with distinct strategies. It might be more appropriate to analyse these groups separately, potentially better in a follow-up publication.

The averaged ratings of the high-pain stimuli appear to be quite variable. Could that have had an influence on the performance?

I am not sure how to evaluate the participants with low prediction accuracies for transition probabilities shown in figure 2. Model comparison in figure 3b shows that the majority of participants can predict the frequency but not the transition probability of the stimuli, despite a preceding training session. The authors may want to consider addressing this in a separate analysis. First, it would be useful to show the relation between rated and generative item frequency, not just transition probability as it is currently done in figure 2. Although this is standard, just treating every participant as equal does not appear right to me. To solve this, I see a few approaches that can be followed separately or combined:

(1) creating subgroups according to the preferred Bayesian model

(2) Between-subjects: correlating the first-level effects with the averaged accuracy scores. This step could be combined with the first step. However, this assumes that all participants process the task in the same way, which appears unlikely.

Was there a reason why some participants were so bad at predicting transition probabilities? They might have expected a sudden change after a series of stimuli of the same intensity? Is there any way to figure this out? The authors should also show the correlation for frequency estimates, preferably side-by-side with the transition probabilities.

Is there a way to explain a few things for the lay reader of the journal? It is not clear to me what some terms mean, e.g. in figure 3: protected exceedance probability etc.

Is there a way to illustrate the different bayesian regressors? It is quite difficult for me to understand how this fitting procedure based on “frequency” and “jump” will look like in a specific trial.

Minor:

There is quite a number of typos in the manuscript.

KL (abbreviated) is introduced in the abstract but not explicitly written.

The low>high (Figure 3 supplement) and the high>low (Figure 4 main document) could be shown in one figure, potentially with warm (high>low) and cold (low>high) colours. However, there might be confusion of the figures: Figure 4 in the main document looks pretty similar to Figure 4 in the supplement.

REVIEWER 1 COMMENTS and REPLY

This manuscript examined the computational and neural mechanisms of temporal statistical learning in the pain experience. The authors showed that participants used Bayesian inference with the dynamic update of beliefs to predict the intensity of upcoming pain, in the absence of any external cues. The different components of the Bayesian modeling, including overall posterior probability (mean), its uncertainty (standard deviation), and model update (consecutive KL divergence) are distinctively related to the activations of classical pain-processing brain regions and additional fronto-parietal and pre-motor regions. Overall, this work is very clear and interesting, and provides novel information to pain neuroimaging researchers. I have a few comments about this manuscript, on which I elaborate below.

1. I'm not fully convinced about the rationale for the frequency modeling. In this study, participants were asked to guess the conditional probability of upcoming pain intensity (i.e., $P(H|L)$ or $P(L|H)$) always. If the participants followed the instruction thoroughly, it is likely that they always tried to infer the underlying transition probability, not the frequency statistics (i.e., $P(H)$ or $P(L)$). The estimation of transition probability also can help prediction because the stimuli sequences were generated based on the dynamically changing transition probability itself. However, the authors set the frequency model as their main hypothesis, and the transition model as a supplementary one. I'm not sure this is the most principled way, considering the experimental design of this study. If the authors expected that some participants mainly use frequency statistics for prediction of pain intensity despite the previous task instruction to estimate transition probability, I think at least there should be some questionnaires to ask participants which strategy they primarily used for prediction in the scanner.

R. We thank the reviewer for such a thoughtful review. We apologize for the lack of clarity in explaining the rationale of our modelling strategy, which we have tried to clarify in the revised manuscript. Inspired by Meyniel's studies with visual stimuli, we aimed to identify whether statistical learning of pain can be described by the following computational principles: (1) is the stimulus sequence learnt using an optimal (or nearly optimal) Bayesian inference strategy, or a non-probabilistic model-free learning rule? (2) does the learning update take into account the volatility of the sequence? and (3) which temporal statistics is inferred, basic stimulus frequencies or transition probabilities? (lines 52-57). We did not have an a priori hypothesis about which temporal statistics participants inferred - it was an open question for us. We did not ask participants to rate the frequency of the stimuli because it can be simply derived from their transition probability ratings; in contrast, transition probability estimates cannot be derived from frequency estimates (lines 85-92). Indeed, if participants are able to predict the frequency, they might not also be able to predict the transition probability of the stimuli; however, if they can predict transition probabilities, they should also have access to the lower-order statistics, such as the frequency of the stimuli.

At the end of the task we asked participants for general comments about the strategy they used. It was not a structured questionnaire, but a debriefing conversation with the participants. The majority of subjects thought that it was difficult to predict the sequence but they were starting to get a sense of the temporal pattern after a while. Some tried to use strategies like counting, but they tended to abandon it to favour a more spontaneous approach of “feeling the flow of the sequence”. Many used a parallel with music, as if they were feeling “the rhythm” of pain. A minority of participants reported to have no clue about the stimulus sequence and were rather bored by the task (lines 338-343).

2. In line with 1, the selected model was based on frequency statistics, but the behavioral results were about the transitional probability. This seems inconsistent to me. Also, the overall correlation between subjective rating and true transitional probability is mediocre, and there were even negative correlations for some participants. Would the Bayesian jump transition model be selected, if the overall correlation was very high? I’m wondering whether the authors can provide some clarification on this.

R. We are grateful to the reviewer for noting this inconsistency - we realise it caused some confusion and we apologise for this. In a revised figure 2, we show the correlation between generative and rated frequencies, as well as transition probabilities (TPs), for each participant. We also report the statistical significance in the main results section (lines 93-113). We note that the sequences were volatile, so inherently rather difficult to predict. The TP model could have won model comparison, at group level, if more subjects had a higher correlation between true and rated TPs. We tried to make this point clearer in the text.

Figure 2. Behavioural results. (a) Relation between generative and rated frequencies, $p(H)$, in each participant (one regression line per participant). (b) Relation between generative and rated transition probabilities, $p(H|L)$, in each participant (one regression line per participant). (c) Estimation accuracy, as measured by the correlation coefficient between generative and rated probabilities: frequency $p(H)$ and transition probability $p(H|L)$; each circle represents one participant. The boxes show the quartile of the data and the whiskers illustrate the rest of the distribution, except for points that are classified as “outliers” based on the inter-quartile range. The two sets of correlations were not significantly different ($z=0.376$, $p=0.707$).

3. I'm wondering whether the predicted transitional probability from the Bayesian jump frequency model actually resembles the subjective rating of transitional probability, like Figure 4 of Meyniel et al., 2015, Plos Comp Biol.

R. The Bayesian jump frequency model predicts frequency, not transition probabilities. We have now plotted the frequency predicted by the Bayesian jump frequency model vs. individual participants in Supplementary Figure 1, similarly to what was done in Figure 4 of Meyniel et al., 2015, Plos Comp Biol. We included statistical analyses of this relationship in the "Quality of fit" section of Supplementary Results.

Fig. S1. Frequency of high pain predicted by the Bayesian jump frequency model vs. human participants. One regression line per participant. The black line is shown only for reference.

4. I think the authors may want to correct some typos.

1) Line 68. Figure 2a -> Figure 1b

2) Lines 86, 89, 93, 94. Figure 2 -> Figure 1

R. So sorry about these errors, thanks very much for flagging them. We have now corrected them and carefully checked the manuscript for typos.

REVIEWER 2 COMMENTS and REPLY

Overall this is a well conducted, methodologically interesting paper examining explicit predictions of upcoming pain. It isn't clear to me, however, how important this is to our understanding of how pain is processed.

R. Thanks for reviewing our manuscript and raising a discussion about the functional significance of the study. The main goal of the pain system is to protect the body from harm, and in order to achieve this goal the brain needs to learn to predict pain. This has been studied in the context of conditioning paradigms with visual/auditory cues that predict pain outcomes. Conditioning tasks with mixed sensory inputs investigate the learning and prediction of *associations* between pain and non-pain cues. They do not allow us to understand how neural predictions are generated purely based on streams of noxious inputs. This is a fundamental gap in our understanding of pain processing, for two reasons. First, clinical pain typically involves long lasting streams of noxious afferent signals to the brain; pain is often not random, but it has *temporal regularities* that characterise its temporal evolution (Foss et al 2006, Hutchings et al 2007). Second, associative learning cannot explain the learning of structures, including temporal structures or regularities as already conceived by Lashley in 1951 (see also Dehaene et al 2015). Our study aims to start filling this gap by investigating how temporal regularities are extracted and learnt from streams of noxious inputs, in absence of sensory cues of other modalities. We confirm Lashley's prediction that associative learning does not explain how people predict the temporal statistics of (pain) sequences. We show that people learn internal models of temporal sequences of pain inputs using optimal Bayesian inference. This is critical to understanding how the brain encodes pain sequences. We reframed the introduction to try to make this point clearer (lines 32-49).

Is it actually common for people to consciously process pain probabilities or is this more likely to be an implicit process?

R. As in other domains, we focused on conscious judgement of the relative overall probability of pain, as opposed to looking at autonomic responses or other physiological measures of pain prediction; this was done to allow direct comparisons between the participant's predictions and ideal observers (Meyniel et al., 2016). We did not test whether statistical learning for pain is automatic but, based on previous work, we expect it to happen spontaneously, without requiring the need to explicitly report stimulus probabilities. Indeed, statistical learning of visual or auditory inputs has been reported in songbirds, primates and newborns (Dong and Vicario, 2020; Kaposvari et al., 2018; Meyer et al., 2014; Saffran and Kirkham, 2018; Santolin and Saffran, 2018). Furthermore, previous studies have shown implicit expectation effects from sequences of stimuli in humans (Atas et al., 2014; Rose et al., 2005; Van Zuijen et al., 2006). [Discussion added at lines 229-237]

Isn't it possible that the cognitive evaluations performed here aren't specific to the pain system and that similar processes would occur attempting to evaluate probabilities of any two different stimuli? While the activations within pain regions might argue against this, we also don't have any indications that the processes being examined have any effect on

the processing of the stimuli, so the relevance to our understanding of pain processing is unclear.

R. Temporal statistical learning is thought to involve both modality-specific and supramodal processes, as previously investigated in visual and auditory processing studies (Frost et al. 2015). In our work, we show that neural activity in sensorimotor regions (S1, S2, posterior insula), which are typically associated with the sensory processing of somatosensory stimuli, is associated with the prediction of pain rate. Previous studies using TSL with visual stimuli reported different neural correlates of TSL, encompassing visual regions (Frost et al. 2015). However, we found that the uncertainty of pain predictions engages a superior parietal region which neighbours a region previously reported to be associated with the uncertainty of visual and auditory predictions, showing supramodal encoding capabilities.

It is an important question for future research whether statistical learning modulates pain perception and behaviour. We don't think this question can be fully addressed in a single study, but this work opens a new exciting avenue of research. In principle, we expect neural predictions emerging from statistical learning for pain to shape pain perception and behaviour, by modulating the cortical response to pain and engaging descending modulatory pathways. Similar modulations have been demonstrated using conditioning paradigms (Buchel et al 2014); for instance, visual cues can modulate even spinal responses to noxious stimuli (Eippert et al 2009). We don't see why pain predictions emerging from sequence learning should not engage similar descending modulatory systems: in principle, they should. If they do, this will show how the brain learning of a pain sequence controls the temporal evolution of pain. This is why this field of work is fundamental to our understanding of pain processing, but it is just too vast for one single paper.

-In the discussion it is stated that "we show that temporal inferences of pain are generated using optimal Bayesian inference". As a minor point, this phrasing confuses a model that approximates how the brain works with how the brain actually works.

R. We agree and we have rephrased that sentence (lines 193-194).

In a larger sense, can we really draw conclusions about how well the model approximates what happens?

R. We think that mathematical models are a useful tool to understand the computations that the brain might use to solve a task. By optimising different models against behaviour, we can find the model that best describes behaviour. The normative approach we used in this work has the advantage of offering testable predictions that can be refuted based on data, and to provide mechanistic links between perception/behaviour and neural activity. An example of a very successful application of this approach involves the discovery that the prediction error in delta rule learning algorithms very closely follows neural activity in the PAG and VTA (Schultz et al. 1997). For a more general discussion of the importance of computational neuroscience methods for pain research, please see Seymour & Mancini 2020 and Seymour 2019. Theories will evolve over time and one day there might be new models that explain our data even better than our current models. This is why we provide all the data openly, so that they can be analysed again in the future, if/when new theories are developed.

The model evaluation statistics seem relative – the chosen model outperforms other models tested, but this doesn't necessarily indicate that it is a good fit overall.

R. That's right. To inspect the quality of fit of the winning model (Bayesian jump frequency model), we evaluated the relation between the frequency $p(H)$ predicted by the Bayesian jump frequency model vs. individual participants (see new Supplementary Figure 1). At group level, the correlation coefficients were significantly above 0 ($r=0.829$, $t(35)=8.629$, $p<0.001$).

Fig. S1. Frequency of high pain predicted by the Bayesian jump frequency model vs. human participants. One regression line per participant. The black line is shown only for reference.

-It isn't clear why the authors predict that encoding of temporal probabilities should be encoded in pain processing regions.

R. We apologise for our lack of clarity. Previous work on statistical learning in other sensory modalities has indicated that statistical learning involves both modality-specific and supramodal brain regions (Frost et al. 2015); visual and auditory cortical regions can encode statistical inferences for their respective modalities, so we reasoned that something similar might occur in somatosensory regions. Based on previous work in Buchel's research group, we also expected the insula to be involved in statistical learning for pain. Using a probabilistic, associative learning task, they showed that the anterior insula is implicated in associative predictions, and the posterior insula in sensory processing of pain stimuli (Geuter et al 2017). We clarified this point at lines 58-66.

-In Figure 2, prediction success is shown for an "example participant" but this participant is clearly not a typical example of prediction success but, rather, one of the most successful (their correlation is nearly 40 points higher than the mean). This gives a false impression of performance on this task.

R. We have redrawn figure 2 to show, for each participant, the relationship between true and rated frequencies (a) and transition probabilities (b). We have also included more analyses about the performance of the task in regards to both frequency and transition probabilities inference (lines 93-113).

Figure 2. Behavioural results. (a) Relation between generative and rated frequencies, $p(H)$, in each participant (one regression line per participant). (b) Relation between generative and rated transition probabilities, $p(H|L)$, in each participant (one regression line per participant). (c) Estimation accuracy, as measured by the correlation coefficient between generative and rated probabilities: frequency $p(H)$ and transition probability $p(H|L)$; each circle represents one participant. The boxes show the quartile of the data and the whiskers illustrate the rest of the distribution, except for points that are classified as “outliers” based on the inter-quartile range. The two sets of correlations were not significantly different ($z=0.376$, $p=0.707$).

References

- Atas A, Faivre N, Timmermans B, Cleeremans A, Kouider S. Nonconscious Learning From Crowded Sequences. *Psychol Sci.* 2014;25: 113–119.
- Büchel, C., Geuter, S., Sprenger, C., and Eippert, F. (2014). Placebo analgesia: a predictive coding perspective. *Neuron*, 81(6):1223–1239.
- Dehaene, S., Meyniel, F., Wacogne, C., Wang, L., and Pallier, C. (2015). The neural representation of sequences: from transition probabilities to algebraic patterns and linguistic trees. *Neuron*, 88(1):2–19.
- Dong, M. and Vicario, D. S. (2020). Statistical learning of transition patterns in the songbird auditory forebrain. *Scientific reports*, 10(1):1–12.
- Eippert F, Finsterbusch J, Bingel U, Büchel C. Direct evidence for spinal cord involvement in placebo analgesia. *Science.* 2009 Oct 16;326(5951):404.

Foss, J. M., Apkarian, A. V. & Chialvo, D. R. Dynamics of pain: Fractal dimension of temporal variability of spontaneous pain differentiates between pain states. *J Neurophysiol* 95, 730-6, (2006).

Frost, R., Armstrong, B. C., Siegelman, N., and Christiansen, M. H. (2015). Domain generality versus modality specificity: the paradox of statistical learning. *Trends in cognitive sciences*, 19(3):117–125.

Geuter, S., Boll, S., Eippert, F., & Büchel, C. (2017). Functional dissociation of stimulus intensity encoding and predictive coding of pain in the insula. *Elife*, 6, e24770.

Hutchings, A. et al. The longitudinal examination of arthritis pain (leap) study: Relationships between weekly fluctuations in patient-rated joint pain and other health outcomes. *J Rheumatol* 34, 2291-300, (2007).

K.S. Lashley (1951). The problem of serial order in behavior. L.A. Jeffress (Ed.), *Cerebral Mechanisms in Behavior: The Hixon Symposium*, Wiley, pp. 112-146.

Meyer T, Ramachandran S, Olson CR. Statistical Learning of Serial Visual Transitions by Neurons in Monkey Inferotemporal Cortex. *J Neurosci*. 2014;34: 9332–9337.

Rose M, Haider H, Büchel C. Unconscious detection of implicit expectancies. *J Cogn Neurosci*. 2005;17: 918–927.

Saffran, J. R., & Kirkham, N. Z. (2018). Infant statistical learning. *Annual review of psychology*, 69, 181-203.

Santolin, C. and Saffran, J. R. (2018). Constraints on statistical learning across species. *Trends in Cognitive Sciences*, 22(1):52–63.

Schultz W Dayan P Montague PR (1997) A neural substrate of prediction and reward *Science* 275:1593–1599.

Seymour, B. (2019). Pain: a precision signal for reinforcement learning and control. *Neuron*, 101(6):1029–1041.

Seymour, B. and Mancini, F. (2020). Hierarchical models of pain: Inference, information-seeking, and adaptive control. *NeuroImage*, 222:117212

van Zuijen TL, Simoens VL, Paavilainen P, Näätänen R, Tervaniemi M. Implicit, Intuitive, and Explicit Knowledge of Abstract Regularities in a Sound Sequence: An Event-related Brain Potential Study. *J Cogn Neurosci*. 2006;18: 1292–1303. pmid:16859415

REVIEWER 3 COMMENTS and REPLY

The study by Mancini and colleagues used a series of electrical stimuli to investigate the prediction of the intensity of painful stimuli in healthy subjects. Most of the participants were able to learn the likelihood of high or low-intensity stimulation. The authors used different modelling approaches to study the underlying cortical processes of this prediction. The general approach has been tested before for the visual domain by a different group.

Therefore, the entire approach has been validated already and the other methods appear sound. The prediction of pain has some behavioural relevance and the investigation can be of utmost significance - not only for the pain researchers but for the entire neuroscience community. I particularly like the idea that the predictions are NOT based on cues; this approximates real-life conditions, where visual cues (red squares or blue triangles etc.) rarely indicate the intensity of upcoming pain. I very much acknowledge the availability of data and code. The results support all conclusions and claims. The work can be reproduced through the provided data and code and meets the standards that can be expected in neuroimaging.

In my view, the manuscript is acceptable as it is. However, a few points could be clarified in order to rule out potential questions from some future readers. This could be done in this or in a subsequent publication.

Suggestions:

There are some inconsistencies regarding the statistical threshold (FDR, FWE). I would suggest using the same method throughout the manuscript.

R. We apologise - this is actually a typo in the caption of former figure 4. We have used the same, conservative FDR threshold of $p < 0.001$ and $Z < 3.3$, throughout.

The prediction accuracy varies substantially across participants and there are apparently two different groups of participants with distinct strategies. It might be more appropriate to analyse these groups separately, potentially better in a follow-up publication.

R. Thanks for raising this point. We have added supplementary analyses to address differences in learning strategy and prediction accuracy. Please see sections “Control analyses with model evidence and prediction accuracy as covariates” and “Individual differences in learning strategy” in the supplemental document. Please see our detailed response to your other comment, pasted below.

[moved from below] *Although this is standard, just treating every participant as equal does not appear right to me. To solve this, I see a few approaches that can be followed separately or combined:*

(1) creating subgroups according to the preferred Bayesian model

(2) Between-subjects: correlating the first-level effects with the averaged accuracy scores. This step could be combined with the first step. However, this assumes that all participants process the task in the same way, which appears unlikely.

R. Thanks for this helpful suggestion, which we have followed. We have added supplementary analyses to address differences in prediction accuracy and learning strategy in the supplemental document. These analyses strengthen our key findings.

Control analyses with model evidence and prediction accuracy as covariates

We conducted three separate fMRI analyses, each adding a different covariate to the generalised linear mixed model presented in the main article (regressors of interest: posterior mean $p(\text{high})$, SD posterior, KL divergence, stimulus intensity). The covariates we tested were:

- (1) the evidence of the Bayesian jump frequency model;
- (2) the prediction accuracy, as indexed by the coefficient of the correlation between the transition probability $p(L|H)$ rated by the subject and the true (generative) $p(L|H)$;
- (3) an alternative measure of prediction accuracy, i.e. the coefficient of the correlation between the stimulus frequency predicted by the subject vs. the Bayesian jump frequency model.

The neural correlates of the mean posterior of low/high pain (probabilistic inference), model update and posterior SD (uncertainty of the inference) were very similar across the different control analyses we conducted (figures S3, S4, S5), and also remarkably similar to those reported in the main article (figures 5-7).

Individual differences in learning strategy

Our model fitting and comparison analyses indicate that there were inter-individual differences in the nature of temporal statistics inferred, using Bayesian inference strategy with dynamic update of beliefs. Whereas 23 participants favoured the inference of the frequency of the stimuli, 12 participants preferred to infer the transition probability (TP) of the stimuli, which yields more accurate predictions. Thus, we conducted additional follow-up neuroimaging analyses to explore these group differences.

First, we divided participants in two groups, according to their preferred inference strategy, defined as the model with the highest evidence (frequency: $n=23$, TP: $n=12$). For each subject, we derived the mean posterior inference, SD posterior and model update (KL divergence between two consecutive posterior distributions) in the jump frequency model and in the jump TP model. After convolving them with a hemodynamic response function, we used them as trial-by-trial regressors for BOLD responses, on each individual, separately for the two models. We then contrasted neural correlates of inference, uncertainty and model update between the two groups (preferred learning strategy: frequency vs. TP), separately for each model (Bayesian jump frequency vs. TP).

We found no significant group differences in the neural correlates of predictive inference (mean posterior of stimulus frequency) and uncertainty (SD posterior of stimulus frequency) in either model. However, the update of the jump frequency and TP models (KL divergence)

was associated with increased activity in the left orbitofrontal cortex in the group that favoured frequency inference than in the group that favoured TP inference (figure S6).

The averaged ratings of the high-pain stimuli appear to be quite variable. Could that have had an influence on the performance?

R. Thanks for this comment. To check this, we evaluated, in each participant, the correlation between the high pain rating (averaged across blocks) and the prediction accuracy (measured by the correlation coefficient between rated and true frequencies and transition probabilities). There was no evidence for a correlation between mean pain intensity and prediction accuracy (frequency prediction accuracy by high pain intensity: $r = -0.175$, $p = 0.337$; $p(H|H)$ prediction accuracy by high pain intensity: $r = -0.178$, $p = 0.305$). This information has now been added at lines 110-113.

I am not sure how to evaluate the participants with low prediction accuracies for transition probabilities shown in figure 2. Model comparison in figure 3b shows that the majority of participants can predict the frequency but not the transition probability of the stimuli, despite a preceding training session. The authors may want to consider addressing this in a separate analysis. First, it would be useful to show the relation between rated and generative item frequency, not just transition probability as it is currently done in figure 2.

R. Thanks for this comment. We now show the relation between true and rated frequency, as well as transition probabilities, in a new Figure 2.

Figure 2. Behavioural results. (a) Relation between generative and rated frequencies, $p(H)$, in each participant (one regression line per participant). (b) Relation between generative and rated transition probabilities, $p(H|L)$, in each participant (one regression line per participant). (c) Estimation accuracy, as measured by the correlation coefficient between generative and rated probabilities: frequency $p(H)$ and transition probability $p(H|L)$; each circle represents one participant. The boxes show the quartile of the data and the whiskers illustrate the rest of the distribution, except for points that are classified as “outliers” based on the inter-quartile range. The two sets of correlations were not significantly different ($z=0.376$, $p=0.707$).

Was there a reason why some participants were so bad at predicting transition probabilities? They might have expected a sudden change after a series of stimuli of the same intensity? Is there any way to figure this out?

R. This is a good question that we can't definitively address in this study, although we can offer some speculation. Predicting transition probabilities involves a higher working memory load than predicting frequency. We know that pain can interfere with working memory (Buhle & Wagner 2010), and some people might be more sensitive to this interference than others.

The authors should also show the correlation for frequency estimates, preferably side-by-side with the transition probabilities.

R. Thanks, this is now shown in Figure 2.

Is there a way to explain a few things for the lay reader of the journal? It is not clear to me what some terms mean, e.g. in figure 3: protected exceedance probability etc.

R. This is explained in the Methods / Model Comparison section: “This comparison produces model frequency (how often a given model is used by individuals), model exceedance probability (how likely it is that any given model is more frequent than all other models in the comparison set), and protected exceedance probability (corrected exceedance probability for observations due to chance) [...]. These values are correlated

and would be considered together when selecting the best fit model.” We replicated this explanation in the caption of figure 3 for clarity.

Is there a way to illustrate the different bayesian regressors? It is quite difficult for me to understand how this fitting procedure based on “frequency” and “jump” will look like in a specific trial.

R. It is helpful to look at figure 2 in Meyniel et al 2016 (see screenshot below), which is now referenced in the methods at line 362-363: *“Panel A shows an example of a sequence in which the statistics change abruptly: the first half, from 1 to 150, was generated with $p(X|Y) = 1 - p(Y|X) = 2/3$, and the second half with $p(X|Y) = 1 - p(Y|X) = 1/3$. In this paper, we consider different hypotheses regarding the inference algorithm used by the brain to cope with such abrupt changes (panel B). Some models assume that a single statistic generates all the observations received (“fixed belief”) while other assume volatility, i.e. that the generative statistic may change from one observation to the next with fixed probability p_c (“dynamic belief”). Models with fixed belief may estimate the underlying statistic either by weighting all observations equally (“perfect integration”), or by considering all observations within a fixed recent window of N stimuli (“windowed integration”, not shown in the figure), or by forgetting about previous observations with an exponential decay ω (“leaky integration”). The heat maps show the posterior distributions of transition probabilities generating the sequence in (A) as estimated by each model. The white dash line indicates the true generative value. The insets show the estimated 2-dimensional space of transition probabilities at distinct moments in the sequence. White circles indicate the true generative values.”* (Source: <https://doi.org/10.1371/journal.pcbi.1005260>)

The only difference between the models inferring transition probabilities and those inferring frequencies, it is simply that the posterior probabilities (θ) are tracking different entities.

(A) Example sequence generated from transition probabilities with one change point

(B) Different integration and inference styles

Minor:

There is quite a number of typos in the manuscript.

R. We apologise for this. We have now carefully revised the manuscript.

KL (abbreviated) is introduced in the abstract but not explicitly written.

R. Thanks for noticing this omission, which we have now rectified.

The low>high (Figure 3 supplement) and the high>low (Figure 4 main document) could be shown in one figure, potentially with warm (high>low) and cold (low>high) colours. However, there might be confusion of the figures: Figure 4 in the main document looks pretty similar to Figure 4 in the supplement.

R. We have now shown both contrasts with different colours (red: H>L, blue: H<L) in Figure 4. For consistency, we have also shown both the mean posterior of p(high pain) and p(low pain) in the new Figure 5 (in pink and green respectively), as it might have been confusing to show them separately as we previously did. Note that the neural correlates of p(H) look partially similar to the neural correlates of the high pain stimuli: this is not an error, but it indicates that several brain regions encoding the intensity of pain are capable of encoding also a probabilistic prediction of the frequency of pain.

Figure 4. Brain responses to noxious stimuli (red: high > low pain stimuli, blue: high < low pain stimuli; FDR corrected $p < 0.001$, colorbar shows Z scores thresholded at $z > 3.3$).

Figure 5. Neural correlates of the mean posterior probability of low pain (green) and high pain (pink) in the Bayesian jump frequency model (FDR corrected $p < 0.001$, colorbar shows Z scores > 3.3).

References

Buhle, J., & Wager, T. D. (2010). Performance-dependent inhibition of pain by an executive working memory task. *PAIN*, 149(1), 19-26.

Meyniel F, Maheu M, Dehaene S (2016) Human Inferences about Sequences: A Minimal Transition Probability Model. *PLoS Comput Biol* 12(12): e1005260.

REVIEWERS' COMMENTS

Reviewer #1 (Remarks to the Author):

The authors have provided very thorough responses to my comments. I'm very pleased with this revision and have no further suggestions.

Reviewer #3 (Remarks to the Author):

The authors have answered all of my questions sufficiently.

I recommend the manuscript be accepted for publication.